# Large Language Models Meet Harry Potter:
# A Bilingual Dataset for Aligning Dialogue Agents with Characters

**Nuo Chen**[†][*] **Yan Wang**[‡§][*] **Haiyun Jiang**[‡]**, Deng Cai**[‡]
**Yuhan Li**[†]**, Ziyang Chen**[‡]**, Longyue Wang**[‡] **and Jia Li**[†§]

[‡]Tencent AI Lab
[†]DSA, Hong Kong University of Science and Technology (Guangzhou),
Hong Kong University of Science and Technology
[†]chennuo26@gmail.com,[§] {yanwang.branden@gmail.com, jialee@ust.hk}
**HPD Benchmark**: https://nuochenpku.github.io/HPD.github.io

## Abstract

In recent years, Dialogue-style Large Language Models (LLMs) such as ChatGPT and GPT4 have demonstrated immense potential in constructing open-domain dialogue agents. However, aligning these agents with specific characters or individuals remains a considerable challenge due to the complexities of character representation and the lack of comprehensive annotations. In this paper, we introduce the Harry Potter Dialogue (HPD) dataset, designed to advance the study of dialogue agents and character alignment. The dataset encompasses all dialogue sessions (in both English and Chinese) from the Harry Potter series and is annotated with vital background information, including dialogue scenes, speakers, character relationships, and attributes. These extensive annotations may empower LLMs to unlock character-driven dialogue capabilities. Furthermore, it can serve as a universal benchmark for evaluating how well can a LLM aligning with a specific character. We benchmark LLMs on HPD using both fine-tuning and in-context learning settings. Evaluation results reveal that although there is substantial room for improvement in generating high-quality, character-aligned responses, the proposed dataset is valuable in guiding models toward responses that better align with the character of Harry Potter.

## 1 Introduction

With the emergence of dialogue-centric Large Language Models (LLMs) such as ChatGPT and GPT-4 (OpenAI, 2023), there has been a growing interest among researchers in exploiting the capabilities of these models to develop open-domain dialogue agents. A particularly exciting and challenging aspect of this pursuit involves aligning the behavior of these agents with a distinct character or individual (Kirk et al., 2023; Salemi et al., 2023).

In this paper, using the Harry Potter series fiction as a prime example, we take a significant leap

| Dataset | Sce. | Att. | Re. | Dy. | Sl. |
|---|---|---|---|---|---|
| PchatbotW (2021) | × | √ | × | × | × |
| PeDialog (2019) | × | √ | × | × | × |
| KvPI (2020) | × | √ | × | × | × |
| P-CHAT (2018) | × | √ | × | × | × |
| WOW (2019) | √ | √ | × | × | × |
| Fri-QA (2019) | × | √ | √ | × | × |
| Focus (2021) | √ | √ | √ | × | × |
| UltraChat(2023) | √ | √ | × | × | × |
| LaMP(2023) | √ | √ | × | × | × |
| **Ours** | √ | √ | √ | √ | √ |

Table 1: Datasets Comparison. **Sce.**, **Att.**, **Re.**, **Dy.** and **Sl.** denote **Scenes**, **Attributes**, **Relations**, **Dynamic** and **Storyline**, separately.

towards aligning dialogue agents with characters in a story. Throughout this process, we discover that despite the assistance of super-powerful LLMs such as ChatGPT and GPT4, we encounter numerous challenges: (1) Firstly, the knowledge of LLMs is primarily dominated by real-world knowledge, which may diverge or even contradict to the character's story setting. In fact, ChatGPT and GPT-4 have developed a substantial understanding of the Harry Potter series and the wizarding world behind it, yet they still occasionally produce hallucinations that contradict the story setting. For instance, GPT4-act Harry might attempt to visit Ron's house using the Hogwarts Express, despite the fact that this train exclusively operates between King's Cross Station and Hogwarts. (2) Secondly, LLMs encounter difficulties in interpreting intricate character relationships, particularly when multiple relationships coexist (e.g., friendships, romantic partnerships and competitors). (3) Finally, the most significant challenge stems from the LLMs' inability to accurately represent the impact of time on characters, leading to potential inconsistencies in their portrayal and development within the story.

The dialogue on the right side of Figure 1 effectively illustrates the difficulties (2) and (3). We all know that Harry and Ron are very close friends, but

---

[*]Equal Contribution. [§] Indicates Corresponding authors.

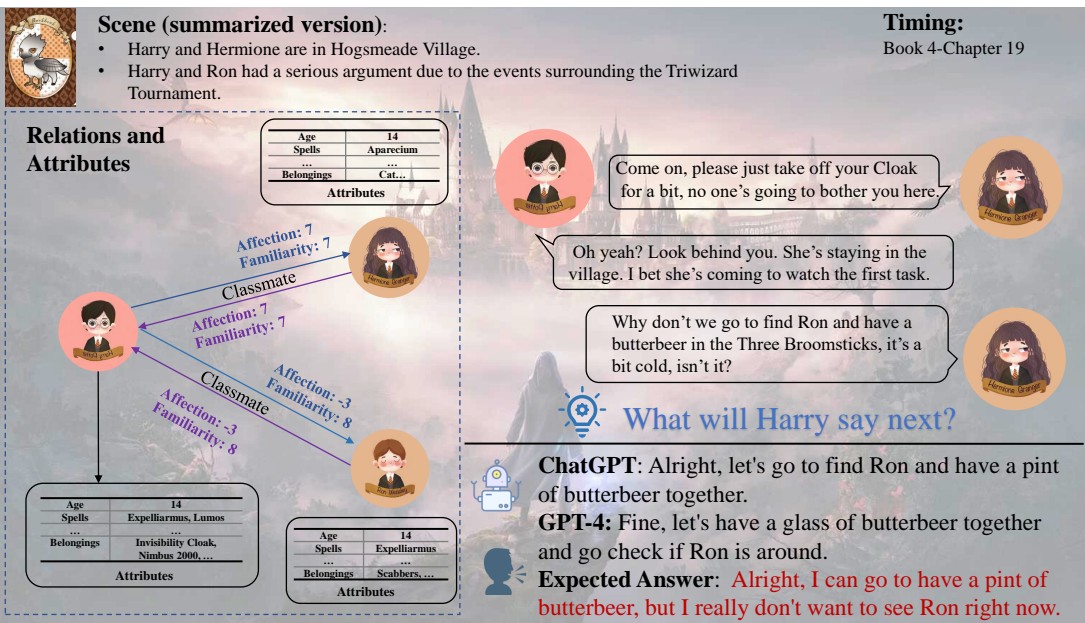

Figure 1: A conversation between Harry and Hermione selected from Book 4-Chapter 19 in the Harry Potter Series. In this example, we present the scene, timing of the conversation, relations and attributes of speakers. Texts in red refer to the expected response.

they sometimes experience conflicts as well, like most adolescents. In Book 4-Chapter 19, Harry and Ron have a heated disagreement over the "Triwizard Tournament" registration (Ron believes that Harry has hidden the truth from him, but in reality, Harry has not), leading to a period of mutual avoidance. At this point, when Hermione suggests to Harry that they should find Ron for a Butterbeer, Harry's normal response would be reluctance to see Ron. However, even after providing an extensive context as a prompt, both ChatGPT and GPT-4 still generate responses that portray Harry and Ron as good friends, which is clearly inconsistent with the situation at that time.

The appearance of the previously mentioned challenges is unsurprising, considering that current dialogue datasets intrinsically lack accurate annotations and guidelines for handling dynamic character relationships and attributes. As shown in Table 1, while some existing datasets (Zhang et al., 2018; Zheng et al., 2019; Jang et al., 2021; Ding et al., 2023) do include basic information about scenes, relationships, and attributes, the issue is that this information remains static and does not change over time. For example, if John and Harry are labeled as "friends", their relationship will remain as friends across all dialogue sessions.

In this paper, we propose Harry Potter Dialogue (HPD), a dataset for facilitating the study of Dialogue Agents - Character aligning. This dataset encompasses all dialogue sessions from Harry Potter novels (English and Chinese versions) [1]. In total, we obtain 1042 dialogue sessions for training (containing 1 positive response only) and 149 sessions for testing (containing 1-3 positive responses and 9 negative responses in average). We also annotate each conversation with essential background information that we believe is useful for aligning dialogue agents with Harry, including **dialogue scenes**, **speakers**, **character relationships**, and **attributes**. With the goal of giving a full picture of the speakers in dialogue, we have labeled each speaker with 12 types of relationships with Harry Potter and 13 types of attributes. Please note that even if the speakers in two different dialogues are identical, their relationships and attributes may change due to the story's progression.

The main contributions of this paper can be summarized into three aspects:

- We introduce the HPD dataset, designed to facilitate the study of aligning dialogue agents with characters. For each dialogue session in the Harry Potter novels, we provide all the background information that we believe may be helpful in assisting models to generate Harry Potter-aligned responses.

- This is a bilingual dialogue dataset that does not rely on machine translation. The only difference between languages is that they are

---

[1]Due to the space limits, we only introduce the **English** experimental results in this paper.

based on Harry Potter novels in different languages. As a result, the data quality in both languages is comparable, making it suitable for investigating the impact of language differences on the task of aligning dialogue agents with characters.

- The experimental results show that HPD can help LLMs, such as ChatGPT, better align with the behaviors of Harry Potter. However, the degree of alignment is still far from the level of human experts, indicating ample room for further improvement.

## 2 Task Definition

We use the Harry Potter novels as our test-bed with the aim of aligning dialogue agents with Harry Potter in a story. The generated responses of such a dialogue agent should be not only relevant to the context, but also seem like something Harry would say at the time and scene.

Figure 1 shows some main factors that affect behaviors of Harry in a conversation. The first factor is the conversation history, which is the most important factor that determines Harry's response. The *scene*, which is the second factor, provides details about the motivation (*Hermione invites Harry to have a butterbeer with Ron in the Three Brromsticks*) of this dialogue. The third factor is the participants' information (**attributes and relations**), obviously, Harry will say very different things to different characters, such as Malfoy and Hermione. The latter two factors belong to the background information and are dynamically determined by the timing of this dialogue (*Book 4-Chapter 19*), and they are continuously varied over the storyline.

Formally, the task of aligning dialogue agents with characters in a story can be defined as follows: Given a dialogue history $\mathbf{H}$, corresponding dialogue scene $\mathbf{S}$ and participants information $\mathbf{P}$ as input, which evolve depending on the development of storyline. The dialogue agent is supposed to generate a response $\mathbf{Y} = \{y_1, y_2, ..., y_n\}$:

$$\mathcal{Y} = \underset{Y}{\operatorname{argmax}} P(\mathbf{Y}|\mathbf{H}, \mathbf{S}, \mathbf{P})$$

$\mathbf{Y}$ is supposed to be not only fluent and natural, but also highly relevant to $\mathbf{S}$ and $\mathbf{P}$.

## 3 Dataset Construction

A high-quality dataset including all pertinent information in Section 2 is the prerequisite for aligning

dialogue agents with characters in a story. Unfortunately, so far there are currently no publicly available datasets that provide information about the dialogue scene and participants. To facilitate the study of this task, we construct a new dataset from the popular fictions Harry Potter Series, in the hopes of creating a Harry Potter-aligned dialogue agent. All dialogue sessions that Harry participates in are collected in our dataset, along with fine-grained annotated dialogue scenes and participant information. We recruit four avid Harry Potter fans (professional annotators, no crowd-sourcing) for the annotation work in this study.

We collect three parts of information to construct our dataset, as shown in Figure 1: 1) The dialogue part (Section 3.1) contains all utterances in the dialogue sessions, as well as the speaker's name of each utterance. 2) The scene part (Section 3.2) includes the summarization of the text around the dialogue session. 3) Finally, the speaker information part (Section 3.3), which consists of attributes and relations of characters, is shown in the left part of Figure 1. Please note that these scenes, attributes, and relations are time-sensitive, they may change as the storyline go on, so we should annotate them session by session.

### 3.1 Dialogue Construction

Dialogue sessions in the books are divided into a training set and a test set. Their main difference is that each training session contains only one positive response, while each test dialogue session consists of at least 1 positive response (human-written ground-truth) and 9 negative responses. We hope this test set may support the evaluation of both dialogue generation and retrieval.

**Training Set** We request the annotators to extract all multi-turn dialogues from the books. Besides, the speaker name of each utterance in the session is labeled as well.

**Test Set** The effectiveness of dialogue models should be evaluated using a well-designed test set. However, if we directly select the test dialogue sessions from the books, it may meet serious knowledge leakage problems, i.e., the fact that we evaluate in the test set also presents in the pre-training corpus. To prevent this problem, we deliberately design a test set in the following steps:

- First, we manually select some raw dialogues that meet the following requirements: (i) Dia-

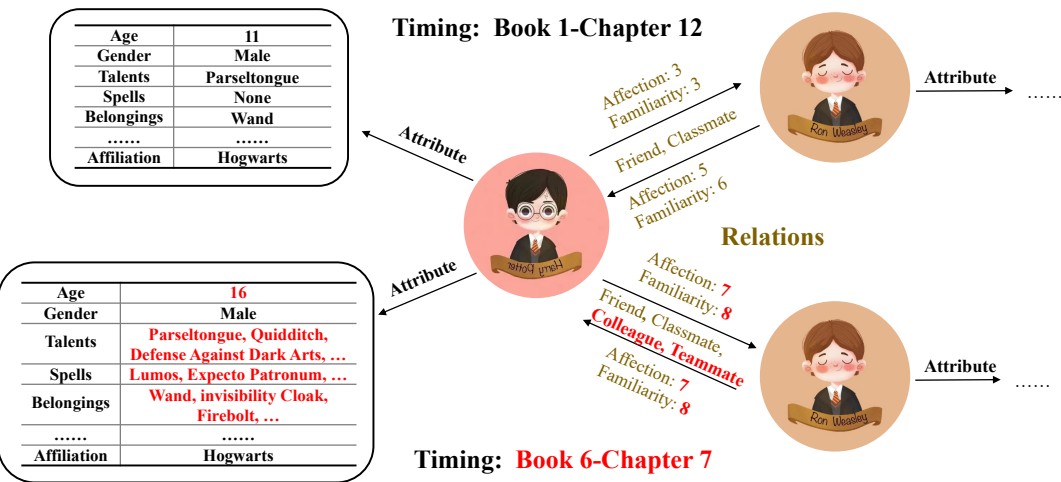

Figure 2: Data examples from two different timings: Book 1-Chapter 12 and Book 6-Chapter 7. Words in color denote the changed relations and attributes.

logues with only one speaker, which contain only one or two sentences; (ii) Dialogues, in which there is no response from other speakers to the last question. For these samples, we pick out dialogues that are relevant to Harry and can be answered from Harry's perspective to construct the test set.

- Second, since directly composing a high-quality Harry-like response is quite challenging for the annotators, we try to alleviate the difficulty with the help of LLMs. Specifically, we utilize ChatGPT and GPT-4 to generate potential responses for each selected dialogue session under in-context learning settings. Subsequently, each model is needed to predict 5 responses as the candidates at least.

- Third, we ask three annotators to select the most reasonable responses from the predictions as the positive response (ground-truth)[2] while the remaining samples are taken as negatives. A holdout annotator is responsible for integrating the annotations. If the selected response is not comprehensive enough or lacks certain information, rewriting is required. Furthermore, if all predictions are considered unreasonable, the holdout annotator would write a response from scratch.

This setup has advantages over employing a single annotator to label predictions: One annotator may regard some predictions as positive responses while the other may label them as negatives. Hence, the holdout annotator can be used to guarantee the

quality of annotated answers when disagreements appear. These strategies alleviate spam and bias, and thus get a high-agreement dataset. Finally, we manually double-check and revise mistakes to further ensure the data quality.

## 3.2 Scenes Construction

In order to offer accurate location information and textual details for each dialogue, we further annotate scenes. We assume that the texts that surround and immediately relate to the dialogue in the novel provide ample scene information. Therefore, we initially instruct the annotators to extract these texts from the novel. Subsequently, to leverage the capabilities of GPT-4, we employ it to summarize the extracted texts. To ensure the utmost precision and accuracy, two skilled annotators are required to meticulously calibrate the summarized texts, resulting in the final scene data.

It is worth noting that we deliberately abstained from directly utilizing the extracted texts as scenes, primarily due to two compelling reasons: Firstly, the inherent length of the original text exceeds the maximum limit imposed by most LLMs, rendering it impractical for our purposes. Secondly, in real-world scenarios, the attainment of such exceptionally high-quality scenes is highly improbable, as scene information typically originates from video captions or rule-based templates.

## 3.3 Attributes and Relations Construction

One of the most important and appealing properties of our benchmark is the fine-grained annotated character information, which includes the attributes and relations of the characters. With the goal of

---

[2]Some questions may have multiple valid answers

providing in-depth and comprehensive character information, we collect 13 fine-grained attributes and 12 relations. We take into account all of these attributes and relationships that are dynamically evolving as the story goes on. Thus, we annotate them chapter by chapter. Considering not all characters are essential to understanding and driving the story in Harry Potter series, we target 113 important characters to annotate their attributes and relations, such as Harry, Ron.

We divide the **attributes** into two categories: (1) inborn; (2) nurture. The former denotes some innate attributes or abilities, which contains {*Gender*, *Age*, *Lineage*, *Talents*, and *Looks*}. The latter refers to properties acquired through efforts, including {*Achievement*, *Title*, *Belongings*, *Export*, *Hobby*, *Character*, *Spells* and *Nickname*} (some cases are presented in Figure 2). In total, we collect 13 attributes for each character, which basically cover most properties in the Harry Potter series.

The **relations** between Harry and other characters can be classified into binary relations and discrete relations. The former include 8 types, which are {*Friend*, *Classmate*, *Teacher*, *Family*, *Lover*, *Opponent*, *Teammate*, *Enemy*}. Multiple binary relations can exist between two characters. Harry and Ron, for instance, are friends, classmates, and teammates in the Quidditch team (In Book-6 only).

In order to fully represent the relationship between two characters, we need to know not only the type of relationship (the binary relations) they have but also their familiarity and affection for each other. We annotate affection and familiarity as 4 types of discrete relations: (1) Harry's *Familiarity* with someone, (2) Harry's *Affection* for someone, (3) someone's *Familiarity* with Harry, and (4) someone's *Affection* for Harry.

The difference between these types can be illustrated by the following two examples: 1) Draco Malfoy hates Harry, but he is also familiar with Harry. So his Affection for Harry is low but his Familiarity with Harry is high. 2) In addition, ever since Harry lost his parents, Dumbledore has shown great concern for Harry. Hence, his Familiarity and Affection for Harry are high, while Harry's Familiarity with Dumbledore is relatively low.

Figure 2 is another example of how attributes and relationships have changed over the course of the story. In Book 1-Chapter 7, Harry has just met Ron on the train to Hogwarts and is an alien in the wizarding world. So his affection and familiarity

Figure 3: Affection Definition and Examples.

Figure 4: Familiarity Definition and Examples.

with Ron are relatively low (1 and 2, respectively), and he isn't aware of any spells. As the story progresses, however, in Book 6-Chapter 7, he is a full-fledged wizard, and Ron is his best friend. So their affection and familiarity are high at this time (7 and 8, respectively). Harry also masters a lot of spells such as *Expecto Patronum* and *Expelliarmus*, and has some equipments such as his broomstick *Firebolt* and the *invisibility cloak*.

**Affection Definition**   *Affection* is rated on 21 levels, ranging from -10 to 10, where -10 and 10 indicate the lowest and highest affection, respectively. A positive affection level means the character has a positive relationship with Harry and vice versa.

Figure 3 shows the detailed definition and some examples of different Affection levels. For example, 1 refers to "*someone meets Harry for the first*

| Statistics | Train | Test |
|---|---|---|
| *per dialogue* | | |
| Average Turns | 13.8 | 7.8 |
| Maximum Speakers | 20 | 8 |
| Minimum Speakers | 2 | 2 |
| *per utterance* | | |
| Average Length | 32.9 | 28.3 |
| Maximum Length | 77 | 26 |
| Minimum Length | 3 | 3 |
| Total Dialogues | 1042 | 149 |

Table 2: Data statistics of collected dialogues.

*time*", when Harry first met Ron and Hermione in Book 1, Harry's *Affection* to both is 1. As for −10, it means the deepest hatred, and the most obvious example is Harry and Voldmort.

**Familiarity Definition** Similarly, we also rate *Familiarity* with a 11-point scale, which ranges from 0 to 10, where 10 is the highest and 0 is the lowest. We present detailed definitions for each *Familiarity* level in Figure 4. Concretely, 0 denotes strangers, and 10 denotes very close friends who stay together for many years and are very familiar with each other's habits, secrets, and temperaments. Ron in Book 7 meets this condition.

**Annotation** During annotation, we ask annotators to annotate Attributes and Relations following the above definitions and examples. To eliminate the effects of individual bias, we ask three of them to repeatedly label the Attributes and Relations chapter by chapter. And a holdout annotator (senior) is responsible for the quality of annotated data when the former three annotators have disagreements. Finally, we manually re-check all annotated data and revise some controversial annotations to further control the data quality.

**Claim** Notice that we hope to provide as rich character information as possible for the community, even if some of them seem redundant in this work. Therefore, we collect 13 types of attributes for each character and 12 types of relations in the collected HPD. We leave plenty of opportunity for other research communities to investigate which information is helpful in their work. For example, it can support other tasks such as the sentiment analysis of Harry Potter. One may not need to include all the fine-grained annotated information in his/her study, because it is still challenging.

## 3.4 Data Statistics

The detailed statistics of dialogues are shown in Table 2. The training set and test set contain 1042 and 149 dialogue sessions, respectively. Of note, we initially collect 1471 dialogues for constructing the training set, and we filter out those dialogues that are without Harry, leading to 1042 conversations for consideration. Most of the conversations involve multiple speakers, with the maximum one including up to 20 speakers. It is obviously a serious challenge to the dialogue agents.

## 4 Experiments

In this section, we conduct extensive experiments to investigate 1) whether Annotations in HPD can assist LLMs in aligning with characters, and 2) if so, whether the degree of alignment can reach human-level performance. We will discuss these questions in section 4.3.

## 4.1 Baselines

We build multiple strong baselines in our experiments, which can be divided into two types: generation-based and retrieval-based systems. Due to the page limit, here we only introduce the generation task in HPD. For details of retrieval tasks in HPD, please refer to Appendix B and D.

**Models** We implement different generative models in two ways: *fine-tuning* and *in-context learning*. For the former, we fine-tune **Alpaca**[3] **(6B)** and **ChatGLM-6B** on our dataset. For the latter, we deliberately design prompts for **GPT3** (Brown et al., 2020) (*text-davinci-002*), **ChatGPT** (*gpt3.5-turbo*), and **ChatGLM** (Zeng et al., 2022) (*chat-glm-130B*). Furthermore, to explore the effect of annotated fine-grained background knowledge, some methods (Alpaca, GPT3, and ChatGPT) are implemented in two different settings: 1) **base setting** with a prompt that only includes task description, one dialogue example, and dialogue history; 2) **rich-persona setting** (denoted as **Per-Model**) with a prompt contains all annotated background information in HPD as in-context learning exemplars. The detailed prompts can be found in Appendix C (Table 8 for base setting, and Table 10 for rich-persona setting).

---

[3]https://github.com/tatsu-lab/stanfordalpaca

| Model | Automatic Evaluation | | | GPT-4 Evaluation | | | Human Evaluation | | |
|---|---|---|---|---|---|---|---|---|---|
| | **Bleu-1** | **Dist.1** | **Rough-L** | **Relv.Sce.** | **Relv.Att.** | **Relv.Re.** | **Relv.Sce.** | **Relv.Att.** | **Relv.Re.** |
| *Fine-tuning* | | | | | | | | | |
| CGLM*-6B | 1.2 | 18.4 | 3.4 | 1.21 | 1.10 | 1.01 | 1.34 | 0.67 | 0.67 |
| Alpaca | 2.0 | **30.2** | 10.8 | 3.37 | 1.19 | 1.51 | 2.03 | 1.19 | 1.51 |
| Per-Alpaca | 12.2 | 20.4 | 13.8 | 3.51 | 1.34 | 1.51 | 4.04 | 1.34 | 1.51 |
| *In-context learning* | | | | | | | | | |
| CGLM*-130B | 14.2 | 25.1 | 15.7 | 6.71 | 6.01 | 6.04 | 7.38 | 6.04 | 5.37 |
| GPT3 | 9.8 | 23.0 | 14.6 | 2.68 | 2.68 | 3.36 | 4.70 | 4.70 | 4.03 |
| Per-GPT3 | 22.6 | 20.1 | 16.5 | 12.75 | 6.04 | 3.36 | 12.08 | 6.71 | 5.37 |
| ChatGPT | 33.1 | 19.0 | 20.2 | 33.56 | 38.92 | 37.58 | 32.89 | 34.23 | 32.89 |
| Per-ChatGPT | **33.6** | 19.4 | **22.4** | **37.58** | **42.95** | **46.98** | **35.57** | **45.63** | **48.99** |

Table 3: A comprehensive evaluation of all baselines on HPD test set. Per-Model means the model with prompts in rich-persona setting. Here we report the percentage of generated responses ranked as the best one for each dialogue session (**top-1 ranking**) in GPT-4 and human evaluation. CGLM refers to ChatGLM. We report the average ranks of these LLMs in Figure 5. **Dist.1** is short for **Distinct-1**.

## 4.2 Evaluation Metric

To comprehensively measure whether the texts generated by various models exhibit a Harry Potter-aligned tone, we employ three different kinds of evaluation methods: **reference-based (automatic)**, **GPT-4 based**, and **human-based**. Although some existing studies (Liu et al., 2023; Peng et al., 2023) claims that GPT-4 is good-enough to assess the quality of generated texts, we found that human judgment is still the most thorough and realistic assessment of whether the generated text is Harry Potter-aligned in our task.

**Reference-based Metrics** We employ **Rough-L** (Lin, 2004), **Bleu-1** (Papineni et al., 2002), **Distinct-1** as our metrics to validate the relevance and diversity of the generated responses.

**GPT-4 Evaluation** We evaluate the persona's consistency based on the three criteria: Relevance with the Scene (**Relv.Sce.**), Relevance with the Attributes (**Relv.Att.**) and Relevance with the Relations (**Relv.Re.**). Considering the potential drawbacks of having GPT-4 to assign scores for all responses individually, such as the risk of confusing GPT-4's comprehension, we have opted for an alternative approach. We instruct GPT-4 to **rank** the generated texts based on above distinct criteria. This ranking process allows us to evaluate the performance of different models effectively. The detailed prompts are in Appendix C, Table 9.

**Human Evaluation** In light of the discovery of some poor GPT-4 annotation cases, we further instructed our annotators to revise and rectify the ranking results of GPT-4 for each test data, leading

to human-evaluation results. During the annotation, annotators were allowed to consult the original text for reference when ranking. Moreover, as long as at least one annotator among the three made adjustments to GPT-4's ranking results, we would adopt the modified results as the human evaluation results. If multiple annotators made revisions to a single result, we would take the average of their modifications as the final human evaluation result.

## 4.3 Results

**Whether the HPD can assist LLMs in aligning with characters?** To answer this question, we present the results of these baselines in Table 3, and observe that across all GPT-4 and human-based metrics, methods perform significantly better in the rich-persona setting compared to their performance in the base setting. This improvement, obviously, should be attributed to the additional background information provided by the HPD.

Additionally, for ChatGPT, the improvement in persona consistency-related metrics, including Relv.Att. and Relv.Re., is more pronounced (for example, compared to ChatGPT, Per-ChatGPT has increased by 11.4% and 16.1% in these two metrics, respectively). This is mainly because, in most cases, the dialogue history has some overlap with the scene, and ChatGPT can be consistent with the scene by merely using the dialogue history. For GPT-3 and Alpaca, their understanding of dialogue history is not as strong as that of ChatGPT, so their improvements in Relv.Sce. are still noticeable.

**Models VS. Human Experts** Intuitively, we may further raise a question: How good are the gener-

| Category | Relv.Sce. | Relv.Att. | Relv.Re. | Ave. |
|---|---|---|---|---|
| WIN | 77.85 | 77.18 | 73.15 | 76.06 |
| TIE | 1.34 | 4.03 | 8.05 | 4.47 |
| LOSE | 20.81 | 18.79 | 18.79 | 19.46 |

Table 4: Comparison between Per-ChatGPT and Human experts. The win/tie/lose proportions of human-written responses on different metrics are reported.

---

**Scene**: Harry and his family visit the reptile house at the zoo, where Harry strikes up a conversation with a boa constrictor from Brazil... Harry is blamed for talking to the snake and causing the incident.
**Dialogue Position**: Book1-chapter2
**Speakers**: Dudley, Harry...Piers
**Dialogue**: Dudley: Make it move...
the snake: I get that all the time.
Harry: I know, Where do you come from, anyway?
the snake: Boa Constrictor, Brazil.
Harry: Was it nice there? Oh, I see — so you've never been to Brazil?
keeper of the reptile house: DUDLEY! MR. DURSLEY! COME AND LOOK AT THIS SNAKE! YOU WON'T BELIEVE WHAT IT'S DOING!
the snake: Brazil, here I come. . . . Thanksss, amigo.
Piers: Harry was talking to it, weren't you, Harry?

---

GPT-3: I was just asking it where it came from. It's not like I was trying to make friends with it or anything.

---

Per-GPT3: I didn't mean to cause any trouble, I was just curious about where the snake came from. I didn't know it would escape.

---

ChatGPT: Yes, I was just curious about where it came from and if it had ever been to Brazil. But Dudley, you shouldn't be so rude to the keeper of the reptile house. He was just trying to show us something amazing.

---

Per-ChatGPT: Yes, I was. But I didn't make the glass disappear or tell the snake to escape. It was just a harmless conversation. I didn't mean for any of this to happen.

---

Annotated Response: **No, I wasn't talking to it. I was just looking at it and reading the sign about where it came from. You two must have imagined things.**

Table 5: Examples of generated responses by baselines. Here, we have omitted the attributes and relationship information between the characters.

ated texts? Are they as good as the ground-truth? To explore this, we further recruit another three annotators to compare the Per-ChatGPT-generated responses with the human-written responses in the test set. The experimental results, as shown in Table 4, are surprising. Contrary to our common sense that ChatGPT has already reached human-level performance in conversations, humans show a clear advantage in the task of aligning with Harry Potter. In all metrics, the proportion of Pre-ChatGPT outperforming or being on par with humans even lower than 30%. As we analyzed in Section 1, the existing LLMs are still far from being able to align with specific characters.

## 4.4 Case Study

As the case in Table 5, the scene of selected dialogue is about *"Harry finds he is able to converse with snakes."* and Harry is facing questioning from Pears, who often bullies Harry with Dudley. He dislikes and even feels somewhat afraid of the Dudley family and Pears. At that time, Harry doesn't know anything about the wizarding world, and hasn't yet known about or gone to Hogwarts. Hence, Harry is likely trying to downplay the situation, keep his secret (he can talk with snakes) and avoid potential conflict with Piers and Dudley. However, all models cannot capture such a complicated relationship and fail to generate real Harry Potter-aligned responses. Most models generate responses that start with "Yes, I was..." or "I was", which are dull and clearly contradict Harry's actual intention.

Additional examples can be found in Appendix E, Table 11. They demonstrate that when dialogues necessitate a deep understanding of the current context or involve a sudden shift in Harry's relationships with other characters, current LLMs' behavior significantly deviates from Harry's actual actions. This is, obviously, far from satisfactory. Although the Harry Potter series is already included in the training corpus of most LLMs (they know the basic information about characters and their world), relying solely on 'next word prediction' may be insufficient to fully understand the nuanced knowledge embedded within the story behind Harry Potter.

## 5 Conclusion

In this paper, we propose a new benchmark named Harry Potter Dataset (HPD) to promote aligning dialogue agents with characters in a story. Unlike existing datasets, HPD not only contains interesting dialogues, but also scenes, character attributes and relations that are dynamically changed as the story-line goes on. It also provides a well-designed test set to facilitate the evaluation of both generation-based and retrieval-based dialogue agents. Results and case studies show that powerful LLMs are still far from human expectations, proving there is ample room for improvement. Generally, HPD offers several open research problems in character aligning, such as *how to build automatic evaluation metrics for personalized text generation*, *exploring effectively prompting LLMs for character aligning in a story*. We hope HPD can play an crucial role in moving through cracking them.

## Ethical Statement

To avoid the potential issue of using Harry Potter novels, we promise the annotated dataset is developed for non-commercial use. Moreover, we only provide the line number and page number of each collected dialogue in Harry Potter rather than the detailed content of each dialogue session. We further supply the script to extract corresponding raw dialogue data from the novels according to the provided line and page numbers, in which the data format is the same as the data examples in Table 11. As for the annotated character attributes and relations, we have our own copyright and will release for research communities.

## Limitations

The main target of this paper is towards building dialogue agents for characters in a story. In this paper, we present a new benchmark named Harry Potter Dialogue (HPD) in the hope of creating a Harry Potter-aligned dialogue agent. The significant feature of HPD is that it contains detailed scenes and fine-grained attributes and relations of each speaker which are dynamically changed as the storyline goes on. More generally, we expect the core idea of this paper can give insights into other research communities that want to build effective person-like chatbots in the virtual world. Our well-designed test sets could even support RLHF training of characterized dialogue agents. Our fine-grained annotated knowledge also can be used to build other tasks such as sentiment analysis and reading comprehension of Harry Potter. Admittedly, the data in the proposed dataset from the Harry Potter Series is restricted to a specific area, that is, Harry Potter Magic World. Considering the high cost of annotation, our character relation annotation work is restricted to Harry Potter. These concerns warrant further research and consideration when utilizing this work to build intelligent person-like dialogue systems in the virtual world.

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

# A Related Work

Recently, building personalized dialogue systems draw a lot of attention from research communities. Aiming for promoting this area, several efforts and benchmarks (Chen et al., 2021; Danescu-Niculescu-Mizil and Lee, 2011; Zheng et al., 2019; Yang and Choi, 2019; Song et al., 2020; Zheng et al., 2020; You et al., 2020; Chen et al., 2023; You et al., 2021; Xu et al., 2022) have been made, demonstrating promising results for endowing personal style into dialogue systems. Some initial efforts (Danescu-Niculescu-Mizil and Lee, 2011) aimed at modeling characters from movies.

Further developments provide personas via two types: implicit and explicit personalization. In the former streams (Kottur et al., 2017; Li et al., 2016; Zhang et al., 2019), each speaker's personality information can be compressed as the persona embeddings. In this manner, the existing issue of these methods makes it hard to explain their effectiveness. For the latter (Wolf et al., 2019; Song et al., 2020), the personal information are provided as: (1) *dense personas*, such as speaker profile or text-described personas; (2) *sparse personas*, including some personality traits. For example, personas from (Zheng et al., 2019) are formulated as key-value pairs: "Age:xx, Gender:xx, Location: xx".

More recently, several efforts (Dinan et al., 2019; Yang and Choi, 2019; Jang et al., 2021; Song et al., 2023) incorporated scenes and relations knowledge into each dialogue session for encouraging more real personalized conversation. Yang and Choi (2019) presented a open-domain question answering dataset excerpted from *Friends Series*, where each dialogue involves multiple speakers and their relations. Jang et al. (2021) proposed FoCus dataset where the customized responses are generated based on the user's persona and Wikipedia background knowledge.

In this paper, our goal is to align dialogue agents with characters in a story, which requires modeling scenes and speaker information that are dynamically changed as the storyline goes on. However, the personality settings of current studies are static, and are not changed with scenes or times changing. Therefore, we present HPD: Harry Potter Dialogue Dataset, aiming for creating Harry Potter-alignd dialogue agent. In detail, we annotate detailed scenes, attributes and relations of each speaker over given dialogues to help the model have a deeper understanding of the dialogue background information.

# B Baselines Setup

In the following, we briefly introduce each model and describe the training and test details.

## B.1 Baselines

**BERT-FP** is a commonly-used strong retrieval-based dialogue system, which devises several post-training objectives. When fine-tuning BERT-FP, given $n$-1 utterances in each dialogue session, the model is required to find the ground-truth response from candidate answers. Concretely, we first post-train BERT-FP in Harry Potter novels and then fine-tune the resulting model in the collected HPD.

## B.2 Experimental Setup

Our experimental settings can be see in Table 6. Notice that, considering the memory cost, we utilize LoRA-tuning (Hu et al., 2021) and prompts don't contain any examples.

# C Prompts

In this section, we present three different prompts in our experiments: Table 8 shows the prompts of the base setting which only include task description, a dialogue example and the input dialogue. Table 9 presents the ranking prompts for GPT-4 evaluation. Table 10 illustrates the rich-persona setting prompts.

# D Retrieval Task

**Automatic Metrics** For evaluating the retrieval-based model, we also employ some common metrics: MAP (mean average precision), MRR (mean reciprocal rank), and P@1 (precision at one). Recall also be considered, which is used as R10@k, which implies that the correct response exists among the top k candidates out of the ten candidate responses.

**Automatic Evaluation** BERT-FP also performs poorly on $P@1$ score (25.9%) and MAP score (46.8%) in the retrieval-based task. The results show the current state-of-the-art retrieval-based model also can not handle the challenge of our benchmark, and thus, there is ample room for future improvement.

| Parameter | Alpaca | Per-Alpaca | ChatGLM-6B | BERT-FP |
|---|---|---|---|---|
| *Batch size* | 4 | 4 | 1 | 32 |
| *Learning Rate* | $1e^{-4}$ | $1e^{-4}$ | $3e^{-5}$ | $8e^{-5}$ |
| *Epoch* | 5 | 5 | 5 | 20 |
| *Max Length* | 512 | 512 | 650 | 256 |

Table 6: Hyper-parameters setup in fine-tuning.

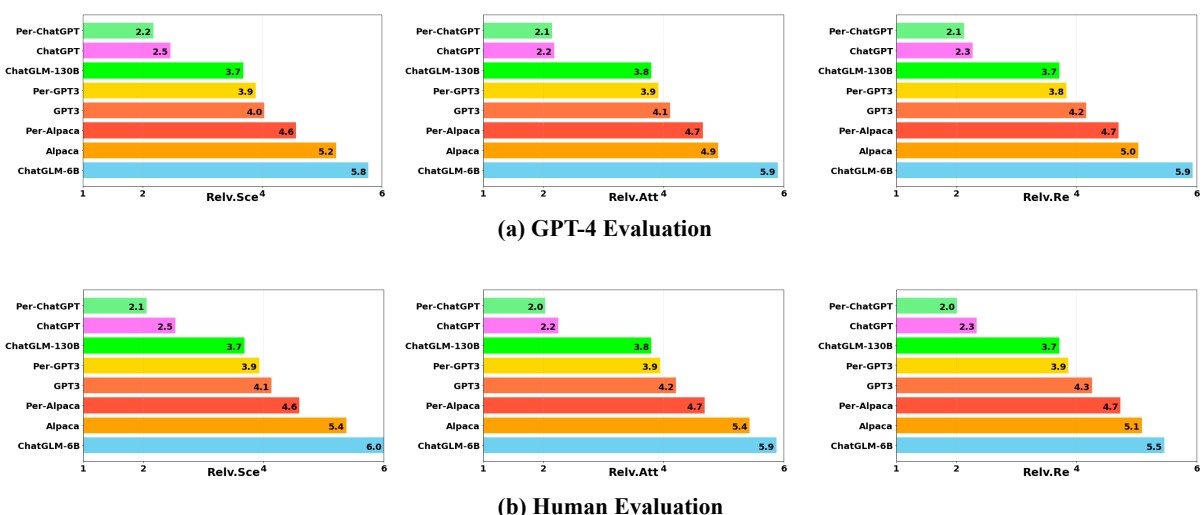

(a) GPT-4 Evaluation

(b) Human Evaluation

Figure 5: Average ranks of generated texts in terms of human evaluation and GPT-4 evaluations.

| | | *Retrieval-based* | | |
|---|---|---|---|---|
| Model | MAP | MRR | P@1 | R10@1 | R10@5 |
|---|---|---|---|---|---|
| BERT-FP | 0.468 | 0.468 | 0.259 | 0.259 | 0.788 |

Table 7: Automatic evaluation results of the retrieval-based model.

# E More Case Study

Intuitively, a higher pursuit of our task is expect the dialogue system can generate the logical response according to attributes and relations between characters. In Chapter 21 of the fifth book, there is a significant scene where Cho Chang presents Harry with a Christmas gift and confesses her feelings. The annotated character relationships and story development depicted in the Table 11 reveal that Harry has harbored feelings for Cho Chang all along. Confronted with Cho Chang's heartfelt confession, it is reasonable to expect Harry to respond with a mix of shyness and excitement, ultimately accepting her advances. This aligns seamlessly with the subsequent progression of the story.

Within this dialogue scenario, it is noteworthy to analyze the responses generated by large models such as GPT3, Per-GPT3 and ChatGPT. They all convey a similar sentiment of reciprocation, expressing an inclination towards liking Cho Chang.

However, it is intriguing to observe that Per-GPT3 and ChatGPT diverge from this pattern and reject Cho Chang's expression of affection. Furthermore, the responses from GPT-3 and ChatGPT-6B are disappointingly concise, failing to adequately capture the nuanced blend of shyness and excitement exhibited by Harry in response to Cho Chang's heartfelt confession.

**Prompts**: Your task is to act as a Harry Potter-like dialogue agent in a Magic World. There is a dialogue between Harry Potter and others. You are required to give a response to the dialogue from the perspective of Harry Potter.
Here is an example:
Dialogue:{ "Petunia: Bad news, Vernon, Mrs. Figg's broken her leg. She can't take him. Now what?",
......
"Vernon: I'm warning you, I'm warning you now, boy — any funny business, anything at all — and you'll be in that cupboard from now until Christmas." }
Harry's Response: **I know, I will obediently obedient, and I won't cause you trouble.**
Keep in mind the following requirements:
Before generating the response, you should read and understand the dialogue content carefully.
```
Input:
Dialogue:   {Input Dialogue}
Onput:
Harry's Response:
```

Table 8: Prompts of the base setting in our experiments.

**Prompts**: You are J.K. Rowling who is the author of the Harry Potter Novels. Here is a scene featuring a conversation between Harry Potter and other characters in Harry Potter Novels. Given 8 potential responses from the perspective of Harry Potter to the scene, you are required to rank the quality of these responses based on the following criteria, respectively:
(1) coherence with relations between Harry and other characters. (short for Coh.Rel);
(2) coherence with Harry's attributes. (short for Coh.Att);
(3) coherence with the scene (short for Coh.Sce).
To help you rank these responses, we additionally provide some background information, including 'Dialogue Position', 'Speaker's attributes' and 'Speakers relations with Harry'.
You should generate the response format with 'Coh.Rel: R8»...»R1; Coh.Att: R8»...»R1; Coh.Sce: R8»...»R1', and then give several sentences to explain your opinion.
```
Input:
Dialogue Position:   {Dialogue Position}
Speakers relations with Harry:   {Speaker relations}
Harry's attributes:    {Harry's attributes}
Scene:   {Scene}
Dialogue:   {Input Dialogue}
R1:   {Response1}
......
R8:   {Response8}
Output:
```

Table 9: Ranking Prompts for GPT-4 in our experiments.

**Prompts**: Your task is to act as a Harry Potter-like dialogue agent in the Magic World. There is a dialogue between Harry Potter and others. You are required to give a response to the dialogue from the perspective of Harry Potter. To do this, you can write out your thought and answer with "Harry's response" at the end.

To better help you mimic the behavior of Harry Potter, we additionally provide the following **background information** of the dialogue:

1. Dialogue position, which represents the timeline of the dialogue in Happy Potter Novels. For example, "Dialogue Position: Book5-chapter28" means the dialogues occurs in Chapter28,Book5.

2. Dialogue speakers.

3. Harry Potter's attributes, which refers to basic properties of Harry Potter when the dialogue happens. It can contains 13 categories: Gender, Age, Lineage, Talents, Looks, Achievement, Title, Belongings, Export, Hobby, Character, Spells and Nickname.

4. Speaker relations with Harry, such as whether he was a friend, classmate, or family member;

5. Harry's Familiarity to the speaker, which ranges from 0 to 10. Concretely, 0 denotes stranger, and 10 denotes close friends who often stay together for many years and are very familiar with each other's habits, secrets and temperaments, where Ron meets this condition in Book 7.

6. Harry's Affection to the speaker, which ranges from -10 to 10. 1 refers to speaker met Harry for the first time. For instance, when Hary first met Ron and Hermione in Book 1, Harry's Affection to them are both set to 1. And -10 means the speaker killed Harry's parents, where Voldemort meets this condition in the novels.

Here is an example:

Dialogue position: Book1-chapter2

Dialogue speakers: Harry, Petunia, Vernon

Harry's attributes:

{"name": "Harry",

"nickname": "The boy who lived",

"gender": "male",

"age": "age 11",

"looks": "Very thin, black hair, emerald green eyes, wearing glasses, knife injury with lightning shape at the forehead",

"hobbies": "None",

"character": "None",

"talents": "None",

"export": "None",

"belongings": "None",

"affiliation": "None",

"lineage": "None",

"title": "The boy who lived",

"spells": "None"}

Speakers relations with Harry: Vernon is Harry's uncle and Petunia is Harry's aunt.

Harry's Familiarity to Vernon: 8

Harry's Affection to Vernon: -4

Harry's Familiarity to Petunia: 8

Harry's Affection to Petunia: -4

Dialogue:{ "Petunia: Bad news, Vernon, Mrs. Figg's broken her leg. She can't take him. Now what?",

......

"Vernon: I'm warning you, I'm warning you now, boy — any funny business, anything at all — and you'll be in that cupboard from now until Christmas." }

Thought: Let's think step by step. According to the conversation history, Vernon warned Harry not to spoil the special day. According to Harry Potter's attributes, he is still very thin, does not know any spells, and has not gone to Hogwarts yet. So he is currently incapable of resisting them. At the same time, based on his affection for them is -4, it means that he relatively doesn't like them, and may even be a little scared. Therefore, Harry possiblely says: I know, I will obediently obedient, and I won't cause you trouble.

**Harry's Response**: **I know, I will obediently obedient, and I won't cause you trouble.**

Keep in mind the following requirements:

1. Before generating the response, you should read the above information and dialogue content carefully.

2. You can not generate the response that is against Harry Potter's attributes and Harry's relations with the speaker.

3. Not every component in the background information may be useful, you should choose some of them to help you generate more concise and comprehensive responses that satisfy the behavior of Harry Potter in the dialogue.

4. Not every speaker have relations, familiarity ad affection to Harry. At that time, you can directly predict what would Harry say only based on the dialogue context..

```
Input:
Dialogue Position:   {Dialogue Position}
Speakers relations with Harry:   {Speaker relations}
Harry's attributes:   {Harry's attributes}
Scene:   {Scene}
Dialogue:   {Input Dialogue}
Output:
```

Table 10: Prompts of the rich-persona setting in our experiments.

**Scene**:Harry stays behind in the common room hoping to receive a Merry Christmas from Cho. When they are alone, Cho starts crying and Harry tries to comfort her. She apologizes and mentions Cedric's death, but Harry tells her that he was good at magic and that Voldemort would have killed him anyway. Cho compliments Harry on his teaching skills and they share a moment under the mistletoe where Cho confesses that she really likes him.

**Dialogue Position**: Book5-chapter21

**Speakers**:Cho Chang, Harry

**Harry's attributes**:

"nickname": "The boy who lived",

"gender": "male",

"age": "15 years old",

"looks": "Very thin, black hair,

emerald green eyes, wearing glasses, knife injury with lightning shape at the forehead",

"hobbies": "None",

"character": "None",

"talents": "Snake cavity, Quidiqi",

"export": "None",

"belongings": "Winter green wood phoenix feathers wand, owl, stealth jacket, sleeve spare mirror, crossbow flying broom, golden eggs, three strong cups, live maps, fake Galon",

"affiliation": "Hogwarts, Dumbledore",

"lineage": "Mixed wizard",

"title": "Boys who do not die, warriors, teacher Dumbledore",

"spells": "Except for your weapons, fluorescent flashes, separation from left and right, calling god guards, bustling, four-point cracking, pouring force, funny bones, fainting to the ground, pointing to me, obstacles, recovery as early"

**Speakers' relations with Harry**:

Cho Chang is Harry's friend and classmate.

Harry's affection to her: 7.0,

Harry's familiarity with her: 4.0,

His affection to Harry: 7.0,

His familiarity with Harry: 4.0

**Dialogue**: Harry: Wha — ? What's up?, Cho Chang: I'm — sorry, I suppose . . . it's just . . . learning all this stuff. . . . It just makes me . . . wonder whether . . . if he'd known it all . . . he'd still be alive . . .,

Harry: He was really good at it, or he could never have got to the middle of that maze. But if Voldemort really wants to kill you, you don't stand a chance.,

Cho Chang: You survived when you were just a baby,, Harry: Yeah, well, I dunno why, nor does anyone else, so it's nothing to be proud of.,

Cho Chang: I'm really sorry to get all upset like this. . . . I didn't mean to . . . Me mentioning Cedric, I know it must be horrible for you, when you saw him die. . . . I suppose you just want to forget about it . . . You're a r-really good teacher, you know, I've never been able to Stun anything before.,

Harry: Thanks,,

Cho Chang: Mistletoe,,

Harry: Yeah, It's probably full of nargles, though.,

Cho Chang: What are nargles?,

Harry: No idea, You'd have to ask Loony. Luna, I mean.,

Cho Chang: I really like you, Harry.?

---

Alpaca: I really like you too, Cho.

ChatGLM-6B: I really like you too, Cho. But I'm not sure if I'm ready for a relationship right now. I'm still dealing with a lot of stuff from the past, and I don't want to bring that into a new relationship. I hope you understand.

ChatGLM-130B: it's okay. I know you're still grieving Cedric's death, and I understand why you're feeling upset. I'm here for you if you need to talk.

GPT-3: I — I like you too.

Per-GPT3: I'm glad I could help, Cho. Cedric was a really great guy and it's hard to lose someone like that. I just try to keep busy and not think about it too much.

ChatGPT: I like you too, Cho. You're a good person and I appreciate your kind words. But I have to be honest with you, I don't think I'm ready for a relationship right now. I still have a lot of things to figure out and deal with. Can we just be friends for now?

Per-ChatGPT: Thanks, Cho. You're a good student and I'm glad I can help you with your spells. As for Cedric, it's not something I can forget easily, but I try not to dwell on it. And about the mistletoe, it's just a silly tradition. Don't worry about it too much.

Annotated Response: **I, um, I really like you too, Cho, I said, feeling my face grow hot. My heart was racing, and I couldn't believe this was actually happening. I just... I didn't know if you felt the same way.**

Table 11: Examples of our generated responses by baselines.