# OpenReview forum: "Large Language Models Meet Harry Potter: A Dataset for Aligning Dialogue Agents with Characters"
_EMNLP/2023/Conference — EMNLP 2023 Findings_

### Official Review · Reviewer_R7XD · 2023-08-04

**Typos Grammar Style And Presentation Improvements:** Rough-L ⇒ ROUGE-L
**Soundness:** 3

**Excitement:**

4: Strong: This paper deepens the understanding of some phenomenon or lowers the barriers to an existing research direction.

**Paper Topic And Main Contributions:**

This paper introduces the 'character alignment' problem; a dialogue agent aligns itself to a specific character by demonstrating its challenges of handling dynamic and multiple character relationships and attributes. To facilitate the study, this paper introduces Harry Potter Dialogue (HPD) dataset. It consists of 1k dialogue sessions obtained from Harry Potter novels. They annotate rich background information such as dialogue scenes, speakers, character relationships, and attributes. Experimental results demonstrate that rich information plays a crucial role in character alignment compared to its baselines only leveraging dialogue history and context.

**Reasons To Accept:**

1. They present a whole-new dataset containing rich annotations to study character alignment.
    1. Especially dynamic and multiple-character relationships have not been explored in the research community.
2. The annotation schemes are designed deliberately, and the experimental results show that the information contributes the better character alignment.
3. The paper reveals that role-playing within a complex context is not a trivial problem for current LLMs.

**Reasons To Reject:**

1. The number of professional annotators for dialogue construction is too small. I also understand crowd workers could not handle this work, but I still have concerns about dataset bias from the small workers.
2. It is not realistic to have well-defined background information for fine-grained character alignment in real scenarios. Thus, the generalizability (Availity) of the dataset is limited.

**Reproducibility:**

3: Could reproduce the results with some difficulty. The settings of parameters are underspecified or subjectively determined; the training/evaluation data are not widely available.

**Reviewer Confidence:**

3: Pretty sure, but there's a chance I missed something. Although I have a good feel for this area in general, I did not carefully check the paper's details, e.g., the math, experimental design, or novelty.

---

> ### Author Rebuttal · Authors · 2023-08-29
>
> Firstly, we sincerely thank you for dedicating your time and effort to review our paper and provide valuable feedback. In response to the concerns raised, we would like to offer the following clarifications:
>
> **Q1**: The number of professional annotators for dialogue construction is too small. I also understand crowd workers could not handle this work, but I still have concerns about dataset bias from the small workers.
>
> >**A1**: Thank you for your thoughtful concerns. We appreciate the importance of addressing the concern about the limited number of professional annotators for our dataset annotations.
>
> Based on our annotation experience, there's a **Trade-Off** between annotation quality and the number of annotators, especially in tasks requiring high expertise and detailed instructions:
>
> *Having fewer annotators allows for better control over annotation quality and inter-annotator agreement through one-on-one training and testing. As the number of annotators increases, providing individualized instruction and testing becomes challenging, potentially leading to lower annotation quality and reduced inter-agreement among annotators.*
>
> In scenarios with high expertise requirements, like annotating our dataset, the quality of annotators takes precedence. Within our project, there is a dedicated researcher responsible for ensuring the data quality and reducing dataset bias:
>
> - **Rigorous Training**: Our annotators are all professionals who are also ardent fans of the Harry Potter series. Prior to annotation, we require them to thoroughly familiarize themselves with the plot of each book in the Harry Potter series. This proactive step significantly improves IAA  and ensures a comprehensive and faithful annotation process.
> - **Trial Annotation**: Following the training phase, we conduct a trial annotation phase. Annotators receive clear guidelines and annotated examples (e.g., Figure 3 and Figure 4 in our main paper). This lets them perform a practice annotation on a smaller dataset, helping us gauge quality. If annotators don't meet the required standards, additional training is given.
> - **Regular IAA Checks**: Annotators receive consistent feedback on their work. For instance, if there is a significant disparity between annotations from one annotator and the rest, a warning and corrective measures are provided. This iterative process ensures prompt identification and rectification of any discrepancies or errors.
> - **Regular Quality Checks**: We uphold stringent quality control by regularly reviewing and evaluating annotators' work chapter by chapter. This ensures consistent, high-quality annotations.
> - **Author's Review**: Finally, as authors, we individually review the annotated content for each book and hold focused discussions on areas that require modifications. This includes addressing inaccuracies in relationship annotations, ensuring comprehensive attribute tagging, and other relevant aspects.  This feedback is then conveyed to the annotators, requesting revisions on the annotated content. This process continues until a unanimous consensus is reached among the authors, with no further modifications needed.
>
> These steps collectively underscore our commitment to maintaining high data quality and mitigating IAA-related issues. We believe that these efforts will contribute to the credibility and reliability of our dataset.
>
> **Q2**: It is not realistic to have well-defined background information for fine-grained character alignment in real scenarios. Thus, the generalizability (Availity) of the dataset is limited.
>
> >**A2**: We appreciate your review and we would like to take this opportunity to respectfully address the concern you raised regarding the clarity and usefulness of our dataset's application.
>
>  In various practical contexts, there is often a demand for intricate background information and prompts. Please allow us to use a gaming example of constructing intelligent language model-driven NPCs to describe the needed background information:
>
> - **Complex Game Numerical System**: In actual gaming scenarios, intricate numerical systems are in place such as Location,  Mission, Records, Level, Experience, Health Points, Magic Points, Attack, Defence, Damage, which is much more complex than the Scene and Relation information exhibited within our paper.
> - **Elaborate Skill and Equipment System**: Taking World of Warcraft as an example, there are 12 roles and 13 races in the game, with each class having over a hundred skills. Additionally, each character can have 18 pieces of equipment, with each piece of equipment having hundreds to thousands of options. This is a system with a complexity far beyond the 'attribute information' of this dataset.
> - **Dynamic Timeline**: In exploration and mystery-solving games, players progress at different rates. Therefore, when  NPCs respond, they must not reveal future information to maintain the players' immersion and experience.
> - **Character Relationship**: This includes affiliations, gender, and more.
>
> All this background information significantly influences dialogues between NPCs and players in specific contexts. Thus, it becomes evident that the annotations in our paper serve as a simulation of real-world application scenarios. Consequently, our labeled data is not "excessively granular" compared to genuine scenarios; rather, it remains "insufficiently granular" by comparison.
>
> Furthermore, please note that we've emphasized our goal of providing as much background information as possible to explore the advancements in **fine-grained character alignment of LLMs** .  Accordingly, we allocate resources to annotate and incorporate potentially beneficial information into our dataset. ***Researchers are afforded the autonomy to select information tailored to their specific needs. If a specific piece of annotation isn't applicable to their potential use case, they're free not to use it.***

---

### Official Review · Reviewer_ZU76 · 2023-08-05

**Soundness:** 3

**Excitement:**

3: Ambivalent: It has merits (e.g., it reports state-of-the-art results, the idea is nice), but there are key weaknesses (e.g., it describes incremental work), and it can significantly benefit from another round of revision. However, I won't object to accepting it if my co-reviewers champion it.

**Paper Topic And Main Contributions:**

This paper introduces the Harry Potter Dialogue (HPD) dataset, which consists of bilingual dialogue sessions from the Harry Potter series, enriched with background information such as dialogue scenes, speakers, character relationships, and attributes. The primary focus is on the task of generating character-specific dialogues, and the results demonstrate that the provided annotations significantly enhance the quality and relevance of the generated dialogues.

**Questions For The Authors:**

A. The dataset only contains train/test but not validation. How the model selection is performed?

B. In scenes construction, when extracting the text surrounding the dialogue, how long is the extracted text? What is the prompt to summarize the text? How long is the summarized text?

C. Does the surrounding text provide enough information to generate appropriate utterances, or does the model need to know more historical information as well? For example, in sec 4.4, lines 520-528 provide more information about the recap and Harry's mental activity, but this information is not included in the scene information of Table 5. Without this information, it's not surprising that models generate utterances starting with "yes, I was…"

D. Why the comparison results between ground-truth and model-generated utterances are surprising? The ground-truth is based on the best one of the five GPT-generated response candidates, meaning at least 80% of the model output would be worse than ground-truth.

E. Why not use the ground-truth selection data to train a scoring model to choose from model-generated output candidates?

F. What is the contribution of each type of annotation to the model improvement?

**Reasons To Accept:**

The paper is well-organized.

The problem addressed in this study is important.

The introduced dataset can be interesting as it includes annotations of dynamic attributes and relationships.

**Reasons To Reject:**

The conclusion that "powerful LLMs are still far from human expectations" is questionable. It's not clear whether this is due to the insufficiency of the provided scene information (Question C) or the lack of careful selection of model outputs (Questions D, E).

The dataset's size is relatively small, and its scope is confined to the Harry Potter series. Some annotation only covers interactions between Harry and others, neglecting relationships involving other characters.

The proposed approach relies on human annotation to improve model performance, which is impractical in real-world applications. A potential solution is to train a model to automatically generate the annotated information.

The paper lacks a discussion of data quality and IAA. The holdout annotator is the major way to solve annotation disagreement, which can introduce that annotator's bias.

**Reproducibility:**

4: Could mostly reproduce the results, but there may be some variation because of sample variance or minor variations in their interpretation of the protocol or method.

**Reviewer Confidence:**

4: Quite sure. I tried to check the important points carefully. It's unlikely, though conceivable, that I missed something that should affect my ratings.

---

> ### Author Rebuttal · Authors · 2023-08-29
>
> Thank you for taking the time to review our manuscript and for providing invaluable feedback. In response to your concerns and questions, we offer the following clarifications:
>
> **Q1**: The conclusion that "powerful LLMs are still far from human expectations" is questionable. It's not clear whether this is due to the insufficiency of the provided scene information (Question C) or the lack of careful selection of model outputs (Questions D, E).
>
> >**A1**: We arrive at the aforementioned conclusion primarily because even with approaches like fine-tuning or in-context learning, rich persona settings of LLMs, such as Per-Alpaca, Per-GPT3, and Per-ChatGPT, fall considerably short of human performance in both automated and human evaluations (see Table 3) during experiments. The comparison between Person-ChatGPT and human experts presented in the Table 4 further reinforces this observation.
>
> Please refer to **A7**, **A8**, **A9** and **A10** for more details.
>
> **Q2**: The dataset's size is relatively small, and its scope is confined to the Harry Potter series. Some annotations only cover interactions between Harry and others, neglecting relationships involving other characters.
>
> >**A2**: Thanks for pointing this out! Allow us to elaborate on our standpoint.
>
> - Firstly, we have considered the interactions among different characters in our annotated dataset, which is evident in the following aspects:
>     - We have applied attribute annotations to all characters.
>     - Initially, we collected a total of 1471 dialogues between characters, from which we selected 1042 as our training set. (Line 397 in the main paper)
>
> Annotated attributes of all characters and the remaining dialogue data have been preserved for future endeavors, which encompasses the modeling of other characters.
>
> - Secondly, it is pivotal to acknowledge that our annotations focus exclusively on the dynamic interactions between Harry Potter and other characters, which is primarily driven by the substantial annotation costs involved.  ***An attempt to annotate interactions between all 113 significant characters would result in a labeling cost and complexity that is 113x112 = 12656 times the current expense, which is unfeasible***. Therefore, the focal point of this paper is to utilize Harry Potter as the subject to delve into the current LLMs' characters aligning's abilities.  If our dataset draws lots of attention from research communities, we are happy to go on annotations to extend other characters.
>
> **Q3**: The proposed approach relies on human annotation to improve model performance, which is impractical in real-world applications. A potential solution is to train a model to automatically generate the annotated information.
>
> >**A3**: We appreciate your review and would like to take this opportunity to respectfully address the concern regarding the clarity and usefulness of our dataset's application.
>
>  In various practical contexts, there is often a demand for intricate background information and prompts. Please allow us to use a gaming example of constructing intelligent language model-driven NPCs to describe the needed background information:
>
> - **Complex Game Numerical System**: In actual gaming scenarios, intricate numerical systems are in place such as Location,  Mission, Records, Level, Experience, Health Points, Magic Points, Attack, Defence, Damage, which is much more complex than the Scene and Relation information exhibited within our paper.
> - **Elaborate Skill and Equipment System**: Taking World of Warcraft as an example, there are 12 roles and 13 races in the game, with each class having over a hundred skills. Additionally, each character can have 18 pieces of equipment, with each piece of equipment having hundreds to thousands of options. This is a system with a complexity far beyond the 'attribute information' of this dataset.
> - **Dynamic Timeline**: In exploration and mystery-solving games, players progress at different rates. Therefore, when  NPCs respond, they must not reveal future information to maintain the players' immersion and experience.
> - **Character Relationship**: This includes affiliations, gender, and more.
>
> All this background information significantly influences dialogues between NPCs and players in specific contexts. Thus, it becomes evident that the annotations in our paper serve as a simulation of real-world application scenarios. ***Consequently, our labeled data is not "excessively granular" compared to genuine scenarios; rather, it remains "insufficiently granular" by comparison***. Therefore, real-world applications may need more complex human annotations.
>
> In terms of training a model for automatic annotation, it's something that's receiving attention in both academia and industries. The intricate labeling in our dataset is hard to replace by model-generated annotations to this moment. Human involvement is definitely still crucial. Nevertheless, we hold the belief that following our meticulous manual annotation process, we can venture into training an annotation model utilizing our curated annotations. This approach would entail a fusion of annotations generated by the model and subsequent manual refinements, with the aim of cutting down the manual annotation effort. This is an avenue we're eager to explore in our future work.
>
> **Q4**: The paper lacks a discussion of data quality and IAA. The holdout annotator is the major way to solve annotation disagreement, which can introduce that annotator's bias.
>
> >**A4**: Thank you for your valuable feedback on data quality and inter-annotator agreement (IAA).
>
> We have thoroughly considered these concerns and integrated the following measures in our paper.  Within our project, there is a dedicated researcher responsible for ensuring and overseeing the quality of the annotation data, who implements the following strategies:
>
>
> - **Rigorous Training**: Our annotators are all professionals who are also ardent fans of the Harry Potter series. Prior to annotation, we require them to thoroughly familiarize themselves with the plot of each book in the Harry Potter series. This proactive step significantly improves IAA  and ensures a comprehensive and faithful annotation process.
> - **Trial Annotation**: Following the training phase, we conduct a trial annotation phase. Annotators receive clear guidelines and annotated examples (e.g., Figure 3 and Figure 4 in our main paper). This lets them perform a practice annotation on a smaller dataset, helping us gauge quality. If annotators don't meet the required standards, additional training is given.
> - **Regular IAA Checks**: Annotators receive consistent feedback on their work. For instance, if there is a significant disparity between annotations from one annotator and the rest, a warning and corrective measures are provided. This iterative process ensures prompt identification and rectification of any discrepancies or errors.
> - **Regular Quality Checks**: We uphold stringent quality control by regularly reviewing and evaluating annotators' work chapter by chapter. This ensures consistent, high-quality annotations.
> - **Author's Review**: Finally, as authors, we individually review the annotated content for each book and hold focused discussions on areas that require modifications. This includes addressing inaccuracies in relationship annotations, ensuring comprehensive attribute tagging, and other relevant aspects.  This feedback is then conveyed to the annotators, requesting revisions on the annotated content. This process continues until a unanimous consensus is reached among the authors, with no further modifications needed.
>
> These steps collectively underscore our commitment to maintaining high data quality and mitigating IAA-related issues.
> We believe that these efforts will contribute to the credibility and reliability of our dataset.
>
> **Q5**: The dataset only contains train/test but not validation. How the model selection is performed?
>
> >**A5**: During the training process, we divided the initial training dataset randomly into training and development sets using an 8:2 ratio. At the end of every epoch, we assessed the loss of the checkpoint model on the development set. Upon completion of training, we chose the checkpoint corresponding to the lowest loss on the development set as our ultimate model.
>
> **Q6**: In scenes construction, when extracting the text surrounding the dialogue, how long is the extracted text? What is the prompt to summarize the text? How long is the summarized text?
>
> >**A6**：Thanks for your suggestion. Our scene summarization encompasses the following key points:
>
> - Our prompts for scene summarization are: ** Below, there will be a scene from the Harry Potter novels along with its chapter reference. Your task is to summarize what happens in the scene within several sentences. Please be comprehensive and fluent.**
> -  The extracted texts for summarizing scenes are consistently composed of multiple paragraphs. Considering the length of dialogues within them, we follow two guiding principles: The first is that the longest texts cannot be longer than one chapter, and the second is that extracted texts can only contain one dialogue session.
> - When utilizing GPT-4 (4k max lengths) to summarize the scene, the maximum output tokens is 500.
>
> **Q7**: Does the surrounding text provide enough information to generate appropriate utterances, or does the model need to know more historical information as well?
>
> >**A7**: Thank you for your insightful question regarding the adequacy of the surrounding text in our paper as the scene for generating appropriate utterances.
> We would like to assure you that we have taken this aspect into careful consideration during the course of our research.
>
> ***In the data collection phase, the length of these summarized scenes is determined by the maximum allowable input length of state-of-the-art models at that time.*** For instance, when we collect the data, ChatGPT has a maximum length of 4k tokens. It's worth noting that other large language models might have significantly shorter maximum input lengths due to memory constraints, like Llama and Alpaca.
>
>  In our experiments, model inputs are beyond just the scene, which includes various components such as dialogues, character relations, and attributes, as well as a timeline (see Table10 and Table 11). **These supplementary elements were meticulously crafted to enrich the context available to the model. However, they may also lead to exceeding the model's maximum input length.** Therefore, we limit the length of summarized scenes in our dataset as we encompass not only the scene but also the conversational dynamics and character interactions in inputs, although sometimes summarized scenes can not convey enough contextual information.
>
> Given the recent advancements in language model capabilities, including the extended input length of up to 16k tokens in ChatGPT, we acknowledge the opportunity to further enrich our scene annotations. We plan to leverage these advancements to improve the depth and breadth of contextual scenes provided to the model.
>
> **Q8**: Why the comparison results between ground-truth and model-generated utterances are surprising? The ground-truth is based on the best one of the five GPT-generated response candidates, meaning at least 80% of the model output would be worse than ground-truth.
>
> >**A8**：Thanks for your careful consideration of our experiments in Table 4.
>
>  **In annotation**, we use vanilla prompts (see Table 8) without any persona and scene information. In annotation, it became evident that the responses generated by these models in this setting were notably subpar, often necessitating subsequent manual revisions.
>
> **In experiments**, prompts of ChatGPT are more complex and with rich personal settings (denoted as Per-ChatGPT in the paper).
>
> ***Thus, their prompts are different.*** The experimental results, contrary to our common sense that ChatGPT has already reached human-level performance in conversations, show a clear advantage in the task of aligning with Harry Potter.  So we argue it is surprising.
>
> **Q9**: Why not use the ground-truth selection data to train a scoring model to choose from model-generated output candidates?
>
> >**A9**：Thanks for your kind suggestion.  In this paper, we mainly focus on the construction of HPD and the validation of some baseline methods.  In our future work, as you've suggested, we plan to delve deeper into more effective training methods for character alignment.
>
> **Q10**: What is the contribution of each type of annotation to the model improvement?
>
> >**A10**: As our paper explains, evaluating model performance most accurately involves manual assessment. ***However, manual evaluation is complex and costly, and ensuring consistent scoring for ablation studies is challenging.*** This is why we don't present ablation study results for each annotation type in our paper.
>
> Still, our analysis uncovers interesting insights. Among all types of annotations, scene annotations clearly have the most impact on model responses. Notably, in specific cases like the one shown in Figure 1, attributes and relations also influence performance.

---

### Official Review · Reviewer_Jmrm · 2023-08-05

**Soundness:** 3

**Excitement:**

4: Strong: This paper deepens the understanding of some phenomenon or lowers the barriers to an existing research direction.

**Paper Topic And Main Contributions:**

This paper investigates the problem of aligning LLMs with character. It mainly introduces a new benchmark dataset(HPD) which is designed to generate and retrieve Harry Potter’s response given a dynamic dialogue history, dialogue scene, and participate information. They conduct instruct-fintuning on open-sourced, in-context learning on closed-sourced LLMs. Both automatic metrics “reference-based automatic metrics and GPT-4 ” and human evaluation are conducted to evaluate the generated dialogue. Experimental result shows that the proposed dataset could help LLMs to align with character, however, it is still far from human expectation.

**Questions For The Authors:**

1. See the reason for rejection.
2. Could you explain more about the scene summarization? What is the window size for summarization? How informative? How is the hallucination problem now even if it is relevant to the scene? Please clarify the metric of relevancy to the scene and limitations of current methods.

**Reasons To Accept:**

It proposed a useful benchmark that may help future research on aligning LLMs with characters (Harry Potter). The annotated dynamic scene, relationship, and attributes seems to provide a good informative framework for the alignment challenges.

**Reasons To Reject:**

1. The annotated dynamic scene, relationship, and attributes seem the key to making LLM aligned, which is heavily engineered.
Take the same Harry Potter Story,  it is unclear how this will perform to extend to other characters. One needs to do heavy annotation again to support a different character. This paper didn't give an efficient way for prompt engineering.
2. Also, it is unclear whether this could also help with arbitrary chat prompts used by arbitrary users in the real world. Because users will not give those informative prompts, which makes the application not quite clear and useful.

**Reproducibility:**

3: Could reproduce the results with some difficulty. The settings of parameters are underspecified or subjectively determined; the training/evaluation data are not widely available.

**Reviewer Confidence:**

4: Quite sure. I tried to check the important points carefully. It's unlikely, though conceivable, that I missed something that should affect my ratings.

---

> ### Author Rebuttal · Authors · 2023-08-29
>
> First and foremost, we appreciate the time and effort you dedicated to reviewing our paper and providing valuable feedback. In response to the concerns raised by the reviewer, we offer the following responses.
>
>
> **Q1**: It is unclear how annotations will perform to extend to other characters. One needs to do heavy annotation again to support a different character. This paper didn't give an efficient way for prompt engineering.
>
> >**A1**: We sincerely appreciate your comments regarding the annotation effort and costs in our paper, we want to clarify our intentions with:
>
> -  Firstly, ****our core motivation is to explore the capabilities of current LLMs in character alignment within stories****. To achieve this, ****we create a benchmark called HPD with detailed annotations, aiding the study of character alignment****. Our experiments show that such finely annotated data significantly improves the model's performance in this regard.  Notably, our paper represents the initial step in character alignment, and we have indeed experimented with various prompting methods. However, the pursuit of more efficient prompt engineering or annotation is reserved for our future work, presenting a challenge we leave to the research community.
>
> - Secondly, please note that we have emphasized our goal of providing as much background information as possible to assist researchers in character alignment (Line 381-392). As a result, we have allocated funds to annotate any information we believe could be useful and have integrated it into the dataset. One of our intentions is to offer insights into what kind of information might be necessary to test and enhance the character-aligning capabilities of LLMs. Therefore, in researchers' own practical scenarios, they can annotate information that suits the application's context as needed. ****It is not essential to carry out annotations at the same detailed level as we have done.****
>
> - Lastly, the cost of annotating our dataset is less than 10k US dollars, a remarkably economical investment even for industrial companies. Notably, this cost is significantly lower than the salary of a researcher (which exceeds 100k US dollars) dedicated to this project. In real-world scenarios, we firmly believe that if such finely annotated data can be leveraged to create a dialogue agent that aligns closely with game NPCs, ****the associated annotation efforts and costs would be negligible for game companies even if we need to extend the heavy annotation to different characters.****
>
> **Q2**: Also, it is unclear whether this could also help with arbitrary chat prompts used by arbitrary users in the real world. Because users will not give those informative prompts, which makes the application not quite clear and useful.
>
> >**A2**: We appreciate your review and we would like to take this opportunity to respectfully address the concern you raised regarding the clarity and usefulness of our dataset's application.
>
> In various practical contexts, there is often a demand for intricate background information and prompts. Please allow us to take the LLM-driven NPCs in a gaming scenario as an example to describe the needed background information at least:
>
> - **Complex Game Numerical System**: In actual gaming scenarios, intricate numerical systems are in place such as Location,  Mission, Records, Level, Experience, Health Points, Magic Points, Attack, Defence, Damage, which is much more complex than the Scene and Relation information exhibited within our paper.
> - **Elaborate Skill and Equipment System**: Taking World of Warcraft as an example, there are 12 roles and 13 races in the game, with each class having over a hundred skills. Additionally, each character can have 18 pieces of equipment, with each piece of equipment having hundreds to thousands of options. This is a system with a complexity far beyond the 'attribute information' of this dataset.
> - **Dynamic Timeline**: In exploration and mystery-solving games, players progress at different rates. Therefore, when  NPCs respond, they must not reveal future information to maintain the players' immersion and experience.
> - **Character Relationship**: This includes affiliations, gender, and more.
>
> All this background information significantly influences dialogues between NPCs and players in specific contexts. Thus, it becomes evident that the annotations in our paper serve as a simulation of real-world application scenarios. ***Consequently, our labeled data is not "excessively granular or informative" compared to genuine scenarios; Rather, it remains "insufficiently granular or informative" by comparison***. That is, real-world applications may need more complex informative prompts.
>
> Furthermore, please note that we've emphasized our goal of providing as much background information as possible to explore the advancement in **fine-grained character alignment of LLMs** .  Accordingly, we allocate resources to annotate and incorporate potentially beneficial information into our dataset. Researchers are afforded the autonomy to select information tailored to their specific needs. If a specific piece of annotation isn't applicable in their potential use case, they're free not to use it. That is, they don't need to use such informative prompts as ours.
>
> **Q3**: Explain more about the scene summarization, including window size and hallucination problem.
>
> >**A3**: Thanks for your consideration of our scene summarization. Our scene summarization encompasses the following key points:
>
> - Our prompts for scene summarization are: ""Below, there will be a scene from the Harry Potter novels along with its chapter reference. Your task is to summarize what happens in the scene within several sentences. Please be comprehensive and fluent.""
> - When utilizing GPT-4 (4k max lengths) to summarize the scene, the maximum output tokens is 500.
> - To address potential issues of hallucination, we employ two expert annotators to inspect and calibrate the summarized scenes, as illustrated in our paper LIne 270-273. However, during our evaluation process, we observe that GPT-4 exhibits minimal instances of hallucination when performing the task of scene summarization.

---

### Meta-Review · Area_Chair_4TAY · 2023-09-19

**Recommendation:** 3

**Metareview:**

The authors present a richly annotated dataset (in English and Chinese) for character alignment based on Harry Potter, aiming to provide a benchmark and also to boost research in this field. The reviewers generally appreciate the importance and usefulness of the dataset as well as clarity of presentation but also raise concerns regarding the amount of effort that would be needed to train a character-aligned system (especially one that is not aligned with Harry Potter), the small size of the dataset, and some concerns regarding annotation. The authors have provided responses to address most concerns and therefore I recommend accepting this paper in Findings.

---

### Decision · Program_Chairs · 2023-10-07

**Decision:**

Accept-Findings

**Comment:**

The authors present a richly annotated dataset (in English and Chinese) for character alignment based on Harry Potter, aiming to provide a benchmark and also to boost research in this field. The reviewers generally appreciate the importance and usefulness of the dataset as well as clarity of presentation but also raise concerns regarding the amount of effort that would be needed to train a character-aligned system (especially one that is not aligned with Harry Potter), the small size of the dataset, and some concerns regarding annotation. The authors have provided responses to address most concerns and therefore I recommend accepting this paper in Findings.